# Structural basis for assembly of vertical single β-barrel viruses

Isaac Santos-Pérez[1], Diego Charro[1], David Gil-Carton[1], Mikel Azkargorta[2], Felix Elortza[2], Dennis H. Bamford [3], Hanna M. Oksanen [3] & Nicola G.A. Abrescia [1,4]

The vertical double β-barrel major capsid protein (MCP) fold, fingerprint of the PRD1-adeno viral lineage, is widespread in many viruses infecting organisms across the three domains of life. The discovery of PRD1-like viruses with two MCPs challenged the known assembly principles. Here, we present the cryo-electron microscopy (cryo-EM) structures of the archaeal, halophilic, internal membrane-containing *Haloarcula californiae* icosahedral virus 1 (HCIV-1) and *Haloarcula hispanica* icosahedral virus 2 (HHIV-2) at 3.7 and 3.8 Å resolution, respectively. Our structures reveal proteins located beneath the morphologically distinct two- and three-tower capsomers and homopentameric membrane proteins at the vertices that orchestrate the positioning of pre-formed vertical single β-barrel MCP heterodimers. The cryo-EM based structures together with the proteomics data provide insights into the assembly mechanism of this type of viruses and into those with membrane-less double β-barrel MCPs.

[1] Molecular Recognition and Host–pathogen Interactions Programme, CIC bioGUNE, CIBERehd, Bizkaia Technology Park, 48160 Derio, Spain. [2] Proteomics Platform, CIC bioGUNE, CIBERehd, ProteoRed-ISCIII, Bizkaia Technology Park, 48160 Derio, Spain. [3] Molecular and Integrative Biosciences Research Programme, Faculty of Biological and Environmental Sciences, Viikki Biocenter, University of Helsinki, P.O. Box 56 Viikinkaari 9, 00014 Helsinki, Finland. [4] IKERBASQUE, Basque Foundation for Science, 48013 Bilbao, Spain. These authors contributed equally: Isaac Santos-Pérez, Diego Charro. Correspondence and requests for materials should be addressed to N.G.A.A. (email: nabrescia@cicbiogune.es)

The seminal observation that the major capsid protein (MCP) fold of icosahedral internal membrane-containing dsDNA bacteriophage PRD1 is similar to that of membrane-less adenovirus led to the proposal that these viruses have a common origin[1]. When additional structures emerged, it was clear that this MCP fold (vertical double β-barrel) and virion architecture is common in many viruses infecting cells belonging to all three domains of life[2–4]. Structure-based viral phylogeny has so far established four lineages but cover more than half of the virus families defined by ICTV[5]. One of these is the so-called PRD1-adeno viral lineage[6,7]. Studies on further ecological niches, mostly from extreme environments, have revealed PRD1-like viruses but with two MCPs[8,9]. The assembly mechanism for this type of viruses has remained obscure so far.

*Haloarcula californiae* icosahedral virus 1 (HCIV-1) and *Haloarcula hispanica* icosahedral virus 2 (HHIV-2) are among the latest discovered icosahedral viruses with two MCPs and they belong to the *Sphaerolipoviridae* family[9]. HCIV-1 MCPs (VP4 and VP7) share 79.3% and 73.0% sequence identity, respectively, with corresponding HHIV-2 MCPs but not with their host recognition complexes[9].

In the current work, we determine the near-atomic structures of HCIV-1 and HHIV-2 by cryo-electron microscopy (cryo-EM). The icosahedrally averaged cryo-EM maps show that the building of the capsid lattice made of vertical single β-barrels relies on membrane proteins interacting with the penton proteins at each apex, on the formation of suitably arranged VP7–VP4 heterodimers and on proteins proximal to the membrane vesicle that organize the VP7–VP4 heterodimers and individual VP7 in pseudo-hexagonal two-tower and three-tower capsomers. We anticipate that these assembly requisites are likely to be universal in membrane-containing viruses with two vertical single β-barrel MCPs and some extendible to viruses with double β-barrel MCPs.

## Results

### Capsid organization of HCIV-1 and HHIV-2 and MCPs' structure.
Using the cryo-EM-derived density maps we built de novo the atomic models of HCIV-1 and HHIV-2 major viral proteins (Fig. 1a left and right, Supplementary Tables 1 and 2 and Supplementary Figures 1 and 2). The icosahedral asymmetric unit (IAU) of both viruses is composed of 12 copies of MCP VP4 and 15 copies of MCP VP7 arranging into pseudo-hexagonal three-tower and two-tower capsomers, numbers (Nos.) 1, 2, 3, and 4, 5, respectively, and one copy of the penton protein (Fig. 1a center; triangulation number for the capsid pseudo $T = 28$ *dextro*). The taller VP4 is composed of two vertical single jelly-rolls, one standing on top of the other, while VP7 consists of a vertical single jelly-roll only (with BIDG and CHEF four-stranded sheets; Fig. 1b). VP4 and VP7 share a high structural homology despite the scarce sequence identity (< 20%) with the corresponding MCPs VP17 and VP16 of *Thermus* phage P23-77 (3.2 and 2.1 Å rmsd, respectively) and with the individual barrels of the double β-barrel MCP of *Pseudoalteromonas* phage PM2, which is the prototype of the PRD1-adeno viral lineage[10,11] (Supplementary Figure 3). All independent 12 VP4 and 15 VP7 subunits within the IAU are almost identical. The first four N-terminal residues of all VP7 subunits except three, localized within the two-tower capsomers (I, L, Y; Fig. 1a center), stretch linearly beneath the base of the adjacent (clockwise) VP4 jelly-roll burying ILE4 and LEU7 and interacting via carbon–oxygen hydrogen bonding of GLY5 and ASN3 with VP4 corresponding residue SER189 (Fig. 2a and inset). This interaction together with the larger interface area that subunit VP7 buries with the clockwise VP4 (~⟨1340⟩ Å²; $n = 12$) than with the anticlockwise one (~⟨760⟩

Å²) defines the heterodimer VP7–VP4 as the capsomer building block (Fig. 1a center, e.g. D with A and not with B, E with C and not with A; Fig. 2b). The VP7 loop (residues 149–154; Fig. 1b left dashed-circle), ~60% buried, latches beneath the VP4 $\alpha_1$ helix (residues 169–175; Fig. 2a black-rectangle and 2b) and in close proximity to the further down VP4 loop (residues 181–184) (Fig. 2 black circle and below inset). The relative angular orientation of the individual β-barrels of the heterodimer VP7–VP4 foresees that of the double β-barrel MCP in PM2 (Supplementary Figure 3b). The N-termini of the three 'unpaired' VP7 subunits in capsomers 4 and 5 are disordered (Fig. 1a center). At the VP7 C-terminus residues 167–174, when ordered, form the short $\alpha_2$ helix (about two turns) (Fig. 1b left). The N-termini and C-termini (GLN4 and LEU232) of VP4 subunits complete the anti-parallel BI β-sheet (Fig. 1b right). MCPs within and across the IAU pack via adjacent strands and connecting loops with some ARG residues forming H-bonds or salt bridges with neighboring subunits (Supplementary Figure 4). These interactions may make up for the absence of other minor proteins cementing the pseudo-hexameric capsomers across their edges within the IAU or across the virus facets as seen in double jelly-roll viruses[7,12].

### Proteins directing the capsomers formation.
Assembly of VP7–VP4 heterodimers into two morphologically distinct capsomers is directed by two different protein species underneath the three-tower and two-tower capsomer bases (Fig. 3a–c). De novo poly-ALA modeling in the HCIV-1 corresponding densities reveals (i) a protein with an α+β-fold beneath the three-tower capsomers (hereafter called Global-Positioning-System-III, GPS-III), and (ii) a bundle of five helices beneath the two-tower capsomers (called GPS-II) (Fig. 3a–c). Both proteins are at 10–15 Å distance from the outer membrane leaflet (Fig. 3b and c and Supplementary Figure 5, top). The density corresponding to GPS-III is stronger and more interpretable near the icosahedral three-fold axis (No. 3 capsomer) than in the capsomers 1 and 2 (Fig. 3a). The quality of the fitting of the HCIV-1 GPS-III model into the corresponding HHIV-2 density implies that the proteins are the same. Moreover, in HHIV-2, the density corresponding to GPS-III protein under capsomer 2 is interpretable with an equivalent model (Supplementary Figure 5). This observation supports that the GPS-III proteins adopt the same fold although the higher flexibility and/or weaker linkage to the membrane as the membrane curvature increases towards the five-fold poles might weaken their density closer to the apices. The GPS-III is off-centered from the pseudo-three-fold axis of the three-tower capsomer (Fig. 3d). This asymmetry provides the order for the register of the VP7–VP4 heterodimers onto the GPS-III during capsomer assembly as each of the three heterodimers display a decreasing contact area (K–Z: ~350 Å²; W–V: ~310 Å²; J–U: ~120 Å²; Fig. 2b, d). GPS-II centers the base of the two-tower capsomers (Nos. 4, 5; Fig. 3c, e). The five-helix bundle GPS-II displays a clear pseudo-two-fold symmetry (perpendicular to the $\alpha_3$ helix axis) that stitches two VP7–VP4 heterodimers and two individual VP7 together (Nos. 4, 5; Fig. 3c, inset). In HHIV-2 the equivalent density is very weak. The recognition of these GPS proteins is challenging although large sidechains are distinguishable occasionally in GPS-III (Supplementary Figure 5, center).

### Vertex proteins.
As previous biochemical data[8,9] proposed VP9 to be a minor protein composing the HCIV-1 capsid, we confronted its primary sequence with the location of the large sidechains displayed by the penton density. VP9 was structurally identified to be the penton protein plugging the apices of the

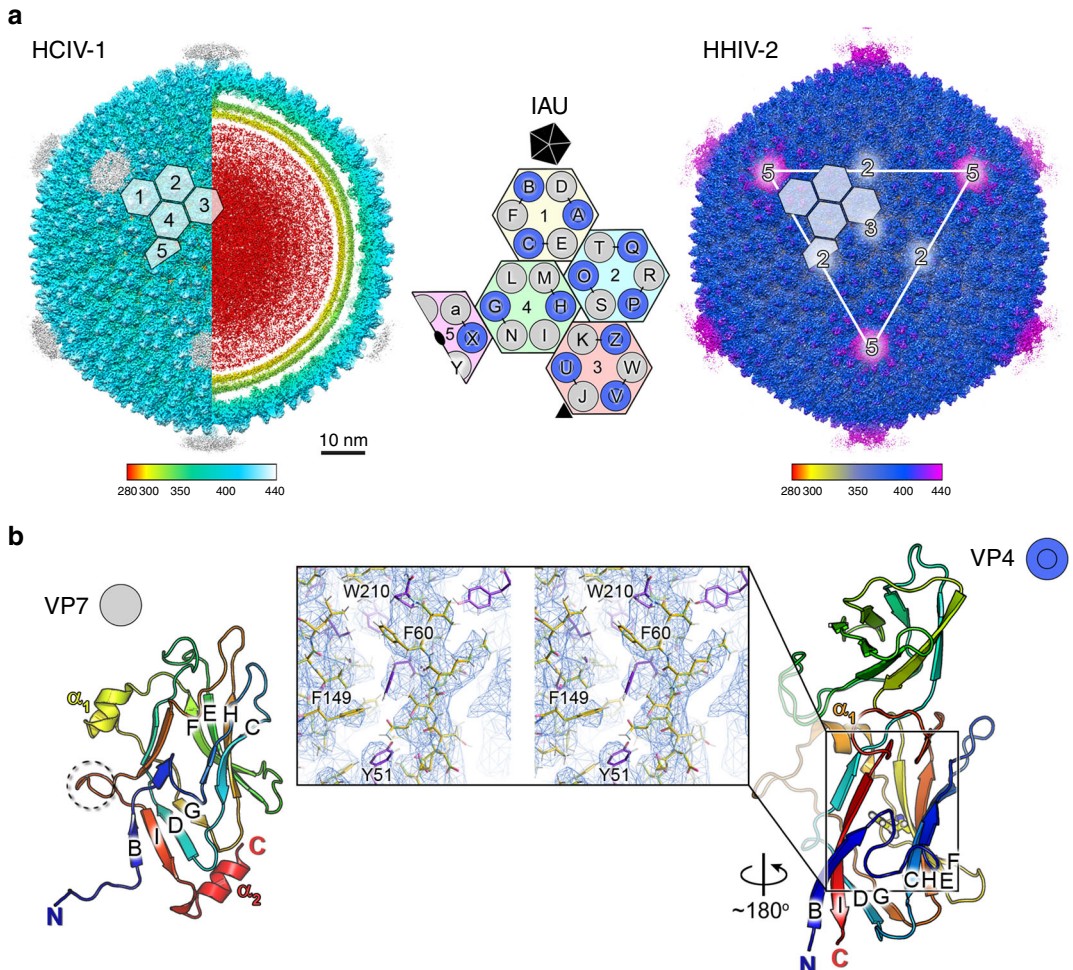

**Fig. 1** Cryo-EM density maps of HCIV-1 and HHIV-2 and MCPs. **a** Overall view of cryo-EM density maps of HCIV-1 (left) and HHIV-2 (right) color-coded by distance from the center (legend below). HCIV-1 is rendered to display the capsid shell (left-half) and the particle interior (right-half), genome in red and inner and outer membrane leaflets in yellow and yellow-lime (vertex complexes have been omitted, see Methods and Supplementary Figures 7 and 8). White-transparent hexagons on top of the capsid densities mark the capsomers (numbered in HCIV-1 Nos. 1–5) forming the icosahedral asymmetric unit (IAU); the white triangle and numbers on HHIV-2 surface mark a facet of the virion and the icosahedral symmetry axes, respectively. At the center, schematic of the capsomers organization with the three-tower capsomers (No. 1, light-yellow; No. 2, cyan; No. 3, pink), composed of three copies of MCP VP4 (blue circles) and three copies of MCP VP7 (light-gray circles) and with the two-tower capsomers (No. 4, light-green; No. 5, light-magenta; the latter sitting on the icosahedral two-fold axis) composed of two copies of VP4 and four copies of VP7; the five triangles composing the black pentagon represents the five copies of the penton protein plugging the vertices. Characters (A–X and a) identify each MCP subunit within the IAU and black short-lines joining the circles identify the VP7–VP4 heterodimers (see Fig. 2). **b** Cartoon representation of the HCIV-1 MCPs VP7 (left) with dashed-circle marking the loop with residues 149–154 [also marked in Fig. 2] and VP4 (right) with residue F149 colored in yellow and represented as stick; inset, stereoview of the region marked by a black rectangle with the VP4 atomic model (as stick and differently colored) fitted into the corresponding 3.7 Å resolution density map (blue mesh contoured at ∼3σ in COOT[26]); some residues including F149 have been labeled

capsid. HCIV-1 VP9 possesses a single β-barrel fold with an N-terminal α-helix (∼34 residues visible) ∼30° tilted relative to the membrane surface (Fig. 4a). About eight residues preceding this motif clamp the base of the adjacent VP9 (Fig. 4b inset left). These N-termini generate a five-helix coiled-coil structure that interacts with the underlying membrane bilayer which accommodates a membrane protein complex (Fig. 4b and inset right). This density is compatible with five spanning transmembrane helices—a structure also used by *Sulfolobus* virus STIV with a double β-barrel MCP[2]—plausibly belonging to five copies of protein VP13 (MW = 8.8 kDa). VP13 was detected by nano-LC–MS/MS analysis of soluble subassemblies clustering with proteins composing the vertex complex (Fig. 4c); its secondary structure prediction (see Methods) suggested VP13 possessing a single transmembrane helix and compatible with the density beneath the penton.

In HHIV-2 the penton protein was identified by gene locus (gene *22*) and the model built accordingly (see Methods); the density below the penton is weaker, indicating a flexibility of the VP9's first 40 residues higher than its counterpart in HCIV-1.

At the HCIV-1 vertices, five copies of the charged VP9 loop connecting strands FE (residues 98–100) fit into the crevice between adjacent VP7–VP4 heterodimers of the peripentonal capsomers (Supplementary Figure 4).

HCIV-1 and HHIV-2 vertex complexes display distinct symmetries (Supplementary Figures 6-8). HCIV-1 possesses a flexible dimeric horn-shaped structure similar to SH1 virus while HHIV-2 has a propeller-like pentameric complex[9,13,14]. The quality of the cryo-EM maps also reveals how the pentons work as adaptors for distinct spikes. In HCIV-1, the center of the penton hosts an extra strand that interacts with the strand G of VP9 and serves as an anchor of the mismatch generated by the

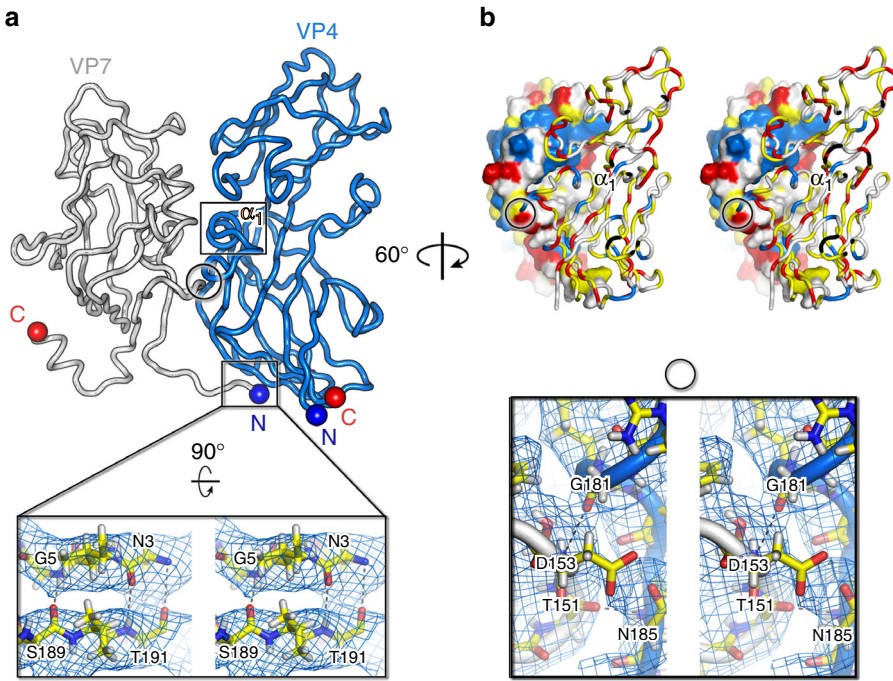

**Fig. 2** VP4–VP7 heterodimer. **a** Heterodimeric VP7–VP4 building block (VP7 in light gray; VP4 marine-blue) with interacting VP7 N-terminal residues. The black-circle marks the VP7 loop, residues 149–154 [indicated also as dashed black circle in (Fig. 1b) left] burying underneath the region underlying the VP4 α1 helix marked by a black rectangle, and contacting the VP4 loop, residues 181-184; blue and red spheres label the corresponding N-terminal and C-terminal. Inset, wall-eye stereoview of the main-chain hydrogen bond interactions (gray dashed lines) between the N-terminal residues of VP7 and residues of VP4 with corresponding density as blue mesh. **b** Top, surface and cartoon tube representation of VP7 and VP4, respectively, highlighting the charge distribution with the hydrophobic residues (A, G, V, I, L, F, M) colored in yellow, residues (R, K, H) positively charged in blue, residues (E, D) negatively charged in red and remaining polar residues in white. Bottom, wall-eye stereoview inset, suitably oriented, showing density as blue mesh and details of main-chain hydrogen bond interactions (black dashed lines) between the residues marked by the black circle in the above panels. Figure was generated in Pymol (https://pymol.org/2/)

dimeric horn (Fig. 4d and Supplementary Figure 7a). The HCIV-1 spike and the pentameric counterpart in HHIV-2 were resolved by localized reconstruction[15] (Supplementary Figures 7 and 8). In HCIV-1, the combination of structural, proteomics, and secondary structure prediction supports VP2 as constituting the horn backbone onto which VP3 and VP6 subunits arrange (Fig. 4c, Supplementary Figure 7 and Supplementary Movie 1). In HHIV-2, the propeller-like spike complex is composed of five copies of putative VP16, a multi-domain protein (four jelly-rolls arranged end-to end) crowning to the below penton. The innermost jelly-roll sits circa orthogonally on top of the VP9 subunits and inserts one of the terminal ends in the inner region of the penton which is coaxially glued together to the central, highly flexible VP2 fiber[14] by an unidentified polypeptide stretch (∼24 residues); the outermost density of the 'blade propeller'—possibly corresponding to VP17—is relatively weak (Supplementary Figure 8).

**Viral assembly**. The specific locations of the three-tower and two-tower capsomers within the IAU indicate that the MCPs assemble onto the membrane vesicle decorated with the GPS proteins. Nano-LC–MS/MS detected VP10 and VP12 (Fig. 4c) and previous genetic and biochemical studies[16,17] showed that the two proteins are the major membrane-associated proteins in both HCIV-1 and HHIV-2. VP10 (aa 173) and VP12 (aa 94) proteins share high sequence similarity across the two viruses (HCIV-1 VP10 and HHIV-2 VP10: ∼57% similarity; HCIV-1 VP12 and HHIV-2 VP12: ∼93% similarity)[14] and secondary structure (PSSpred in I-TASSER[18]) and transmembrane (TMHMM v. 1.0, http://www.cbs.dtu.dk/services/TMHMM/) predictions confidently indicate VP10 with high α-helical content

and possessing a likely C-terminal transmembrane helix whereas VP12 being composed of two transmembrane helices (aa 13–35 and 50–68). These observations support VP10 as candidate for the GPS-II proteins (Fig. 3c).

The presence of membrane protein(s) at the five-fold (the outer-leaflet of the vesicle is reliable up to ∼13–10 Å resolution and appearing increasingly fenestrated towards the five-fold; see Fig. 4b, Supplementary Figure 1d, 9a) and the proteomics results showing stronger affinities of MCPs with vertex proteins (Fig. 4c) endorse the five-folds as the nucleation sites for assembly. These apical membrane complexes—VP13 homopentamers with the transmembrane helices forming a left-handed coiled-coil (Fig. 4b and Supplementary Figure 9b)—act as docking rafts for the VP9 subunits. The pentons in turn would nucleate the attachment of VP7–VP4 heterodimers guided by the presence of GPS proteins that direct and staple VP7–VP4 heterodimers and monomeric VP7 into a pseudo-hexagonal lattice. The absence of ordered proteins connecting neighboring pseudo-hexamers would ease the adjustments or repositioning of the above heterodimers and monomers whose interactions would rely on interface affinities and outermost connecting loops within the MCPs (Supplementary Figure 4).

Cryo-EM image classification also identified genome-devoid particles (procapsids), likely result of defective dsDNA packaging (Supplementary Figure 9c). The 3D procapsid map showed an intact capsid and an unexpanded icosahedral membrane vesicle (inner facet-to-facet diameter of ∼544 Å versus the ∼566 Å of the mature virion; Supplementary Figure 9d). The ∼15% membrane's volume expansion similar to that observed in PRD1 upon genome packaging[19] (a packaging vertex most probably also exists in these

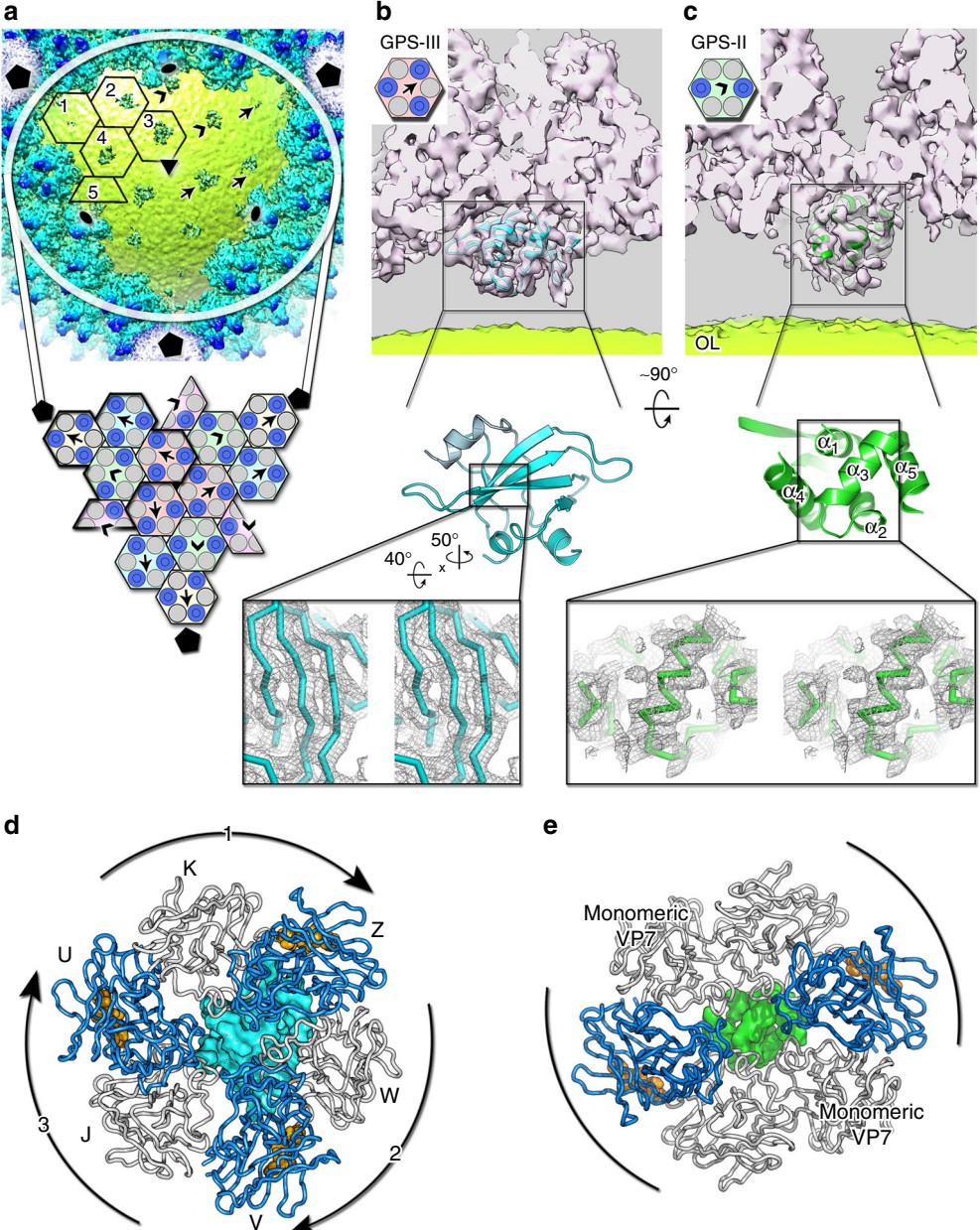

**Fig. 3** GPS proteins beneath the capsid shell. **a** Top, cut-through density of a virus facet viewed along the icosahedral three-fold axis; arrows and arrowheads indicate the density (shades of blue) beneath the three-tower (Nos. 1–3, see Fig. 1a center) and two-tower (Nos. 4 and 5) capsomers (black hexagons), respectively. The outer leaflet (OL) of the membrane is in lime-yellow. Below, schematic of the capsomers composing the facet with one IAU outlined by a thicker black line (represented as in Fig. 1a). Pentagons, triangles and ovals mark the icosahedral symmetry five-fold, three-fold, and two-fold axes. **b** Top, side-view of the Gaussian filtered (1.4 Å width in Chimera[35]) electron density (white 40% transparency) corresponding to the three-tower capsomer No. 3 close to the icosahedral three-fold axis (as from schematic, top left corner) with further density beneath marked by a black rectangle with the GPS-III poly-ALA model fitted in (cyan cartoon) and in lime-yellow the blurred membrane OL; the same atomic model at the center shows an orthogonal view of the GPS-III protein with an inset of a stereoview of the density (gray mesh contoured at ~2σ in Pymol) corresponding to the resolved strands (black rectangle). **c** As **b** but corresponding to the density of the two-tower capsomer No. 4 (as from schematic, top left corner) with the additional density at its center as marked by the black rectangle corresponding to the five-helix bundle GPS-II protein (green cartoon). **d** Top view along the pseudo-three fold axis of MCPs composing the three-tower capsomers as cartoon tube color-coded as Fig. 2a with the off-centered GPS-III protein represented in cyan surface; labels identify the VP7–VP4 subunits and curved black arrow with numbers the putative order of docking/registering of the VP7–VP4 subunits onto the GPS-III. **e** As **d** but for the two-tower capsomers where the GPS-II (green surface) is centered with the $\alpha_3$ spanning the central cavity of the two-tower capsomers stapling together the opposite VP7–VP4 heterodimers (curved black lines) and leaving space for the docking of monomeric VP7

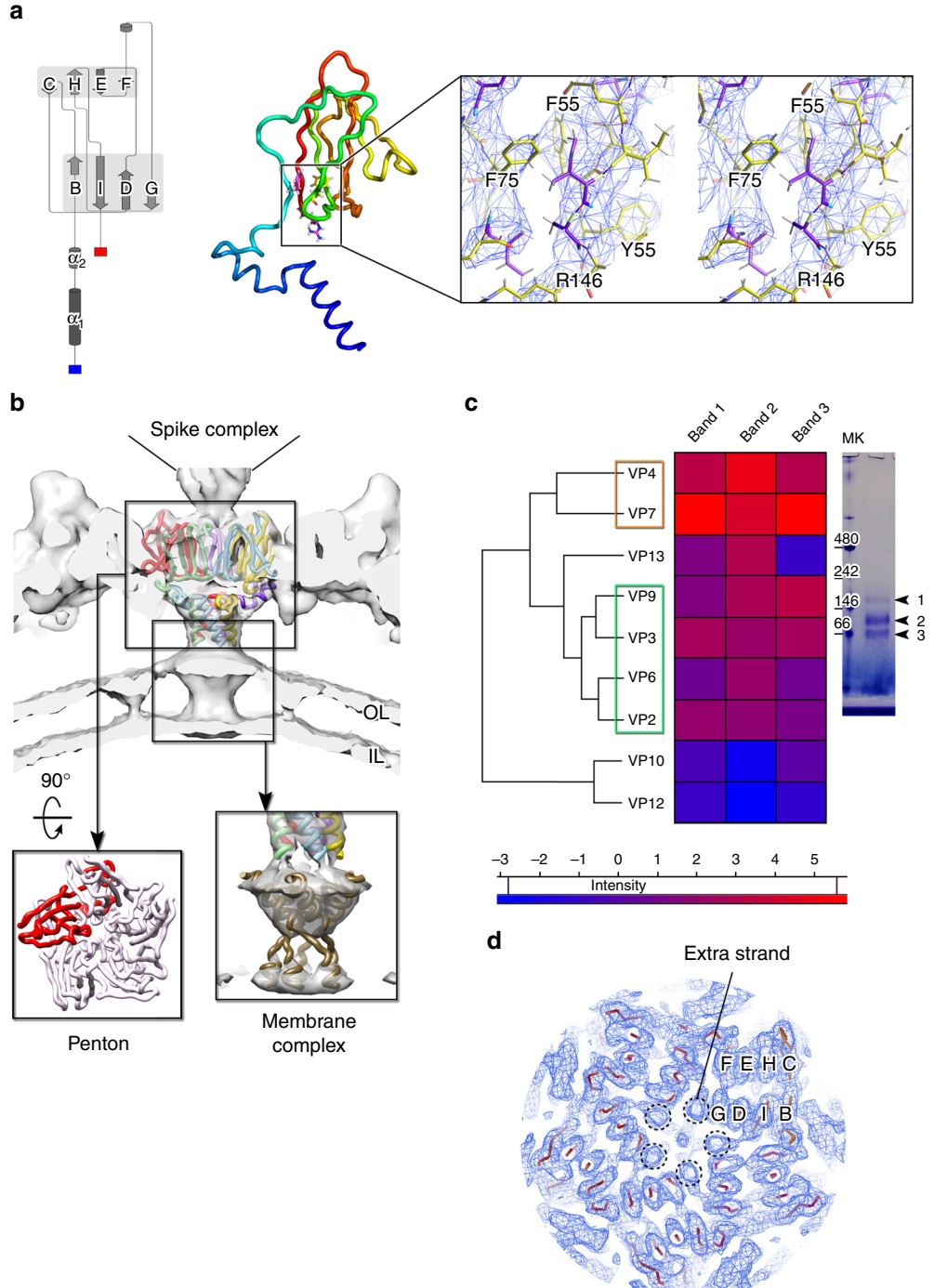

**Fig. 4** Interactions at the vertices. **a** Left, protein structure cartoon of HCIV-1 VP9 penton protein with the labeled jelly-roll strands; center, cartoon-tube representation of VP9 (rainbow coloring: N-terminus, blue; C-terminus, red) with inset showing a stereoview of the corresponding electron density region (as in Fig. 1b inset) marked by the black rectangle with in stick the residues F55, F75, Y55, and R146. **b** Five copies of VP9 models (yellow, magenta, cyan, green, and red) fitted into the corresponding HCIV-1 cryo-EM map (solid white and Gaussian filtered with 2.8 Å width) viewed perpendicular to the icosahedral five-fold axis. Inset on the left, view of the penton complex from the top (one VP9 subunit in red tube, the remaining four in white); inset on the right, the density region corresponding to the membrane protein plugging the membrane vesicle five-fold apices (contoured at higher threshold) and compatible with five copies of the C-terminal transmembrane helix of VP13 (brown tube; see below, Methods and Supplementary Figure 9a-b).
**c** Heatmap depicting the hierarchical clustering (dendrogram on the left) where rows indicate the viral proteins detected according to their normalized abundance factor (NSAF) across the columns which represent the gel bands (arrowheads 1, 2, 3) cut from the representative native gel on the right (MK, markers in kDa). Color scale represents the intensity values as defined as $I = \log_2$ (NSAF) with blue = low abundance, red = high abundance. The orange and green rectangles mark, respectively, the clustering of the MCPs and the proteins composing the vertex complex with identified VP13 grouping closer to the vertex complex whereas proteins VP10 and VP12 are almost equidistant from the two major clusters. **d** View of the cryo-EM map as blue mesh (contoured as in Fig. 1b inset) viewed along the five-fold symmetry axis with fitted the Cα model of the penton in red and with one VP9 jelly-roll labeled as in **a** left; the dashed black circles mark the extra β-strands deriving from the above spike complex and that interact with the VP9 G strands

two viruses), underlies a general virus maturation mechanism in which the increased proximity of the membrane's outer leaflet to the capsid secures the final lock between the two architectural components.

## Discussion

Icosahedral viruses with two vertical single β-barrel as MCPs have posed a conundrum since their discovery[8]. The known mechanisms for virus assembly could not explain the formation of an icosahedral capsid composed of 900 copies of one vertical jelly-roll (e.g. VP7 in HHIV-2), and 720 copies of another vertical jelly-roll (e.g. VP4 in HHIV-2) organized in pseudo-hexagonal, close-packed, capsomers but displaying distinct morphologies (e.g. two-tower and three-tower)[10,13,14]. By investigating the archaeal, halophilic, internal membrane-containing icosahedral viruses HCIV-1 and HHIV-2 at 3.7 and 3.8 Å resolution, respectively, using high-resolution (HR) cryo-EM and proteomics analysis we have finally deciphered the principles governing the organization of vertical single β-barrel viruses. These are based upon: (i) the specific interactions between the N-terminal end of VP7 with residues at the base of the VP4 jelly-roll that define the functional arrangement of the VP7–VP4 heterodimer; (ii) the penton proteins VP9 engaging with homopentamers of membrane protein VP13; (iii) the VP7–VP4 heterodimers interacting with membrane-proximal proteins GPS-II and GPS-III whose location, fold, and oligomerization state dictate the positioning and formation of the pseudo-hexagonal two-tower and three-tower capsomers onto the membrane vesicle. Further, we shed light on the structural motifs (e.g. strand–strand interactions) that lock the architecturally diverse spike complexes into the corresponding pentons illustrating the 'universal adaptor role'[14] of the penton.

In conclusion, our studies elucidate how nature has solved the complexity of arranging vertical single β-barrels and provide insights into the evolutionary consequences of the fusion event of the two consecutive MCP genes, which ultimately has led to the membrane-less vertical double β-barrel MCP assembly seen in adenovirus.

## Methods

**Virus production and purification**. HCIV-1 and HHIV-2 were grown on *H. californiae* (ATCC 33799) and *H. hispanica* (ATCC 33960), respectively (plate lysates and liquid cultures), and purified as described by using polyethylene glycol–NaCl precipitation and rate zonal and equilibrium ultracentrifugation in sucrose[16,17].

**Cryo-EM data collection**. For HR data collection we proceeded in two steps. First, sample and cryo-grid optimization were performed using the in-house EM facilities. The HHIV-2 sample (4 μl at 0.9–1.2 mg/ml) pipetted onto 200-mesh Quantifoil R 2/1 holey-carbon grids was vitrified using a Vitrobot (Mark III—FEI). The HCIV-1 sample at 1.0–1.2 mg/ml was vitrified using a similar procedure. A few grids of each set were pre-inspected as a 'quality control' step on a JEM-2200FS/CR (JEOL, Ltd.) electron microscope, operating at 200 kV at liquid nitrogen temperature before being cryo-shipped for HR imaging to large EM facilities in northern Europe[20].

2,786 cryo-images of HHIV-2 in movie format (7 frames) were collected at the Netherlands Center for Electron Nanoscopy (NeCEN) on a Titan Krios II equipped with a Cs-corrector and a Falcon II direct detection camera (FEI). Images were recorded within a defocus range from 0.7 to 3.0 μm with a dose of ~35 e⁻/Å² at a nominal magnification of ×60,000, producing a final pixel size of 1.34 Å. Subsequently, HCIV-1 data were collected at the electron Bio-Imaging Centre (eBIC) at Diamond Light Source (UK). 3,218 HCIV-1 cryo-images in movie format (27 frames) were recorded on a Falcon II camera within a defocus range from 0.6 to 3.9 μm with a total dose of 36 e⁻/Å² at a nominal magnification of ×59,000, producing a final pixel size of 1.40 Å (Supplementary Tables 1 and 2).

**Image processing and 3D reconstruction**. Movie pre-processing was performed using the MotionCorr software[21] with the inclusion of 6 out of the 7 frames collected for HHIV-2 and 26 out of the 27 frames for HCIV-1, respectively. CTF was calculated using CTFFIND3 and CTFFIND4 for HHIV-2 and HCIV-1,

respectively[22]. Particle picking and extraction for both viruses was carried out using RELION[23] with a box size 760 × 760 pixels for HHIV-2 and 768 × 768 pixels for HCIV-1. The choice of the above box sizes optimized the intensive computing processing time but excluded part of the corresponding spike complexes. These were later determined by the localized reconstruction method[15].

For HCIV-1, 4,584 particles were extracted and initially run through 2D classification into two, four, and ten classes. In the test-run with 10 classes, one class (84 particles) displayed a virion class-average with an empty membrane vesicle with a reduced diameter compared to the remaining nine ones (Supplementary Figure 9c top). Particles of the remaining nine classes were merged together (4,500 particles) and then 3D classified into two classes using the HHIV-2 HR map first determined (see below) but filtered to 60 Å as reference model. The class with 3414 particles was then further 3D refined until convergence. The final HCIV-1 map was post-processed in RELION-3 using the Ewald sphere correction[24,25], a threshold mask and an user applied B-factor of −40 Å² resulting in the gold-standard Fourier Shell Correlation (FSC) for the capsid estimated in 3.74 Å resolution (4.25 Å resolution for the whole virus; Supplementary Figures 1a and 2a, and Supplementary Table 1).

For the HHIV-2 image processing, a total of 14,877 particles were extracted, visually inspected and 3D classified in three classes using as initial reference the low-resolution cryo-EM HHIV-2 map[14]. The 11,446 particles composing the largest class went into auto3D-refinement in RELION[23,24] until convergence. Two circular masks were concomitantly applied during 3D classification and Euler angles refinement excluding the regions of inner radius $R_{in}$ = 338 Å and outer radius $R_{out}$ = 920 Å, respectively; defocus was corrected per particle. Post-processing in RELION-3 using the Ewald correction[24,25], a threshold mask and a B-factor of −40 Å² led to the final HHIV-2 map with a resolution of 3.78 Å for the capsid as judged by the gold-standard FSC (3.89 Å for the whole virus; Supplementary Figures 1b and 2b, and Supplementary Table 2).

Once the HCIV-1 map became available a 3D-refinement was also performed on the minority class composed of particles devoid of the DNA and with the unexpanded membrane vesicle (procapsid); in this case the images were binned. Post-processing was performed as previously explained for the larger class and led to a map at ~15 Å resolution as shown by the gold-standard FSC at the 0.5 criterion (Supplementary Figure 1c). This HCIV-1 procapsid map was used as search model (no mask applied) for a posteriori 3D classification of the HHIV-2 particle population which also identified a very small fraction of DNA-devoid virions (53 particles; Supplementary Figure 9c bottom).

The entire processing of the unbinned images for both viruses was carried out on the Picasso node (Malaga University) of the Spanish Supercomputing Network (https://www.res.es/).

**3D reconstruction of spike complexes**. The structures of the spike-complexes of HCIV-1 and HHIV-2 were determined using the localized reconstruction software[15]. The same virus particles that contributed to the 3D reconstructions were re-extracted from the original micrographs but using a larger box size of 1056 × 1056 pixels so that the full region corresponding to the spike complexes could be subboxed.

To study the HCIV-1 spikes two approaches were used. For the analysis of the structural flexibility, the subboxed 40,968 HCIV-1 spikes underwent to 2D classification using three classes. Two out of the tree classes were merged (56%; 22,804 particles) based on the number of side views present. This class was 2D re-classified in five classes and the three cleanest ones merged, summing-up to a total of 12,034 particles. Finally, these particles were 2D classified in 10 classes (Supplementary Figure 7b).

For the 3D reconstruction of the HCIV-1 spike, the 40,968 extracted vertex complexes were used to generate 3D starting model with C2 symmetry imposed (Supplementary Figure 7a). The resulting map, filtered to 60 Å resolution, was used to 3D classify (with C2 symmetry) the full set of particles into four classes. This produced two spurious classes (48%) and two classes with spikes displaying different apertures, 24% spike complex-1 (closed) and 28% spike complex-2 (open), respectively. After 3D refinement of the individual classes, in addition to the overall FSC resolution estimations of the corresponding maps (threshold masks used), due to the apparent spike flexibility the local-resolution was also estimated in RELION[23] (Supplementary Table 1, Supplementary Figure 7a–c and Supplementary Movie 1).

For HHIV-2 we performed a 3D classification of all particles (137,352 particles) in three classes (no alignment) which informed on the spike occupancy (42% full spike, 22.5% fiberless spike, 35.5% empty pentons). The class with full spike (58,176 particles) was split in two random datasets for 3D reconstruction and postprocessed. The resulting map with the local-resolution estimation is shown in Supplementary Figure 8 (for the FSC estimation, see Supplementary Table 2). The corresponding densities, when possible, were further interpreted in terms of atomic models (see below).

**Model building and refinement**. The different structural components of the viruses were built in density using COOT[26]. First, the atomic models of the HCIV-1 VP7 and VP4 MCPs were manually built aided by β-barrel core templates derived from the crystal structures of P23-77 MCPs[10]. Poly-ALA full models were generated and the geometry optimized via restrained refinement in COOT against

the corresponding individual model-maps. To this end the individual averaged maps of VP4 and VP7 (across the IAU) were initially used but soon after substituted with the individual density of VP4 and VP7 (subunits A and D, Fig. 1a center). Once the poly-ALA models exhibit a satisfactory geometry, they were refined in PHENIX[27] using phenix.real_space refinement with secondary structure restraints 'on' and the corresponding protein sequences put in register. The refinement was then restarted using Rosetta software[28] and finalized in PHENIX v1.13. The quality of the electron density of HCIV-1 at the penton also allowed to manually build the Cα backbone and to register the sequence (from residues 44 to 146) thus identifying it as the VP9 protein (gene 27—Uniprot code: A0A1C7A3R7). Moreover, further density at the N-terminus was clearly visible and attributable to an α-helix —this was modeled as poly-ALA (residues 16–43). Concomitantly, due to the high sequence identity between VP4 proteins and VP7 proteins across the two viruses, the HCIV-1 models were analyzed in the context of the HHIV-2 individual corresponding densities. The HHIV-2 penton protein was first identified by gene locus (gene 22—Uniprot code: H9AZX8) and confirmed by fitting and refining into density as above; the first 40 residues were disordered (20.7% sequence identity with HCIV-1 VP9). At this stage the density below the capsomers of HCIV-1 and HHIV-2 was interpreted. In HCIV-1, the higher quality of the electron density beneath the three-tower capsomer close to the icosahedral threefold axis allowed for the manual Cα trace of the polypeptide that displayed an α+β topology (GPS-III proteins; we assumed a single polypeptide chain despite one breakage in connectivity). The density below the two tower capsomers was also readily interpretable in terms of secondary structure elements as it was mainly constituted by five α-helices (GPS-II proteins). While the N → C directionality of the GPS-III Cα ALA model could be postulated, it was more challenging with GPS-II; both poly-ALA models were fitted in density using phenix.real_space_refine with secondary structure restraints and a conservative weight (no significant match was obtained by structural interrogation of refined GPS-II using DALI server).

The achieved resolution of the spikes in HCIV-1 did not allow to manually build the Cα model and therefore the electron density map was interpreted in terms of secondary structure prediction and modeling with I-TASSER[18] (Supplementary Figure 7c–e). In HHIV-2, the cryo-EM map corresponding to the propeller-like region was generally compatible with jelly-roll-rich domains (Supplementary Figure 8a). However, only the innermost density displayed discernible individual β-strands, which were interpreted by building ab initio ALA-models in COOT moving upwards from the penton (black rectangle in Supplementary Figure 8a and b). Although the connectivity between the three built jelly-roll domains remains unclear (the corresponding densities also become weaker moving outwards), this modeling exercise accounts for circa 210 residues which would indicate the whole protein possibly corresponding to VP16 (253 residues), including the fourth domain compatible with a jelly-roll fold (Supplementary Figure 8b and c). The most external density, corresponding to the 'blade' of the propeller appears to be composed by further domains, two of which display a β-barrel fold oriented as almost capping the VP16 fourth domain (oval in Supplementary Figure 8a, c). This density might correspond to VP17 (418 residues). From the center of the propeller stems the long fiber composed by VP2[14] (Supplementary Figure 8a). Five copies of a non-identified polypeptide stretch glues the central fiber with the penton and the five copies of VP16 (Supplementary Figure 8b and c).

All individually refined protein models were used to generate the IAUs of the corresponding virus maps. The atomic models in the context of the IAU were refined with phenix.real_space using rigid-body, global minimization and adp, with non-crystallographic symmetry (NCS) as constraints across equivalent subunits, with secondary-structure restraints and with conservative weights (Supplementary Tables 1 and 2). Then, the models for the full virus particles were generated, the residues at the boundaries of the IAU inspected and a final rigid-body refinement of the IAU with 60-fold NCS constraints was performed. Refinement and validation statistics, including the EMRinger[29] score for side-chain model-to-map fit, were extracted from the final full virus models (Supplementary Tables 1 and 2).

**Nano-LC–MS/MS and VP13 EM density assignment**. HCIV-1 sample (7.5 mg/ml protein in buffer containing 1 M NaCl, 70 mM MgCl$_2$, 20 mM KCl, 1 mM CaCl$_2$, 50 mM Tris–HCl, pH 7.2 to a final volume of 45 μl) was incubated 30 min at RT, and filtered (Vivaspin® 500 centrifugal filter 1000 kDa MWCO; Beckman Coulter Microfuge 22R F241.5P fixed-angle rotor, 12,000 rpm, 7 min). The flow-through was collected for BN-PAGE and tryptic digestion. Briefly, the sample was run using a NativePAGE system (Invitrogen). Gradient bis–Tris gels (4–16% acrylamide) were used for the resolution of soluble subassemblies of HCIV-1. Gels were stained with a home-made Coomassie blue, and bands were subsequently excised and washed with ultrapure water. In-gel-digested peptides were resuspended in 10 μl 0.1% formic acid and sonicated for 5 min prior to separation by online nanoliquid chromatography (nLC). Peptides were analyzed by electrospray tandem mass spectrometry (MS/MS), using a nanoACQUITY UPLC system (Waters) connected to an LTQ Orbitrap XL ETD mass spectrometer (Thermo) and data was further acquired and processed as aforementioned[30]. Searches were performed using the Mascot Search engine (www.matrixscience.com, Matrix Science) on Proteome Discoverer v.1.4 software (Thermo). Spectra were searched

against a HCIV-1 UniProt/Swissprot database (2017/09). For reliable protein identification, at least two peptides were required to pass the $p < 0.01$ filter (high confidence). Normalized spectral abundance factor (NSAF) was calculated as reported elsewhere[31]. Briefly, protein relative abundance was inferred from spectral counting (defined as the total number of spectra identified for a protein in a sample) where the number of spectral counts (SpC) identifying a protein are divided by protein's length (L), and further divided by the sum of SpC/L for all N proteins in that particular sample. Therefore, relative abundance values are expressed as the % of the total normalized signal that corresponds to each protein. These calculations were carried out over the data obtained in each of the three bands detected in a total of four native gels prepared with samples derived from two different virus preparations. Average NSAF values were calculated for each protein in each gel band, and these values were visualized in a hierarchical clustering heatmap using Perseus software[32]. This analysis identified three viral transmembrane proteins: VP13 (Uniprot: A0A1C7A3R4; 8.8 kDa), VP12 (Uniprot: A0A1C7A3R0; 9.8 kDa), and VP10 (Uniprot: A0A1C7A3R6; 17.7 kDa), with VP13 clearly present in bands 1 and 2 (Fig. 4c). The clustering of VP13 closer to the proteins composing the vertex complex together with VP13 possessing a transmembrane helix at its C-terminus (from 60 to 77 residues; http://phobius.sbc.su.se/index.html and secondary structure prediction I-TASSER[18]; Fig. 4c and Supplementary Figure 9b top left) led to the hypothesis that VP13 might constitute the membrane complex below the pentons. A similar structural role has been assigned to STIV A55 protein[2]. Whether to the existence of structurally well-described single-pass transmembrane proteins with transmembrane helix–helix (TMH) interactions forming left-handed pentameric coiled-coil a bibliographic search was launched. While homodimeric TMH–TMH are relatively common, less so homopentamers with phospholamban (PLN), an integral membrane protein of the sarcoplasmic reticulum, the only one to our knowledge, to have been investigated in detail[33,34]. Alignment of the above sequences showed in the corresponding transmembrane regions a remarkable similarity (Supplementary Figure 8b bottom). In particular pairwise alignment between the TMH of VP13 and PLN showed a sequence similarity of 61% (and identity of 22% and no gaps) much higher than those found for the other transmembrane helices in VP10 and VP12 (data not shown). Further, although at low resolution, the fit of the pentameric atomic model of PLN (PDB ID 2M3B; https://www.rcsb.org/structure/2M3B) into the viral density strikes for its spatial matching of the helices footprint onto the membrane inner leaflet (Supplementary Figure 8b right) supporting the candidacy of VP13 to this location.

**Reporting summary**. Further information on experimental design is available in the Nature Research Reporting Summary linked to this article.

## Data availability

All cryo-EM maps and corresponding PDB models have been deposited in the Electron Microscopy and Protein Data Bank: EMD-0174 (HCIV-1 full map) and PDB ID 6H9C (HCIV-1 full virus model); EMD-0050 (DNA-devoid HCIV-1 map); EMD-0073 (HCIV-1 spike complex-1/open); EMD-0072 (HCIV-1 spike complex-2/closed); EMD-0172 (HHIV-2 full map) and PDB ID 6H82 (HHIV-2 full virus model), EMD-0131 (HHIV-2 vertex complex). The authors declare that all other data supporting the findings of this study are available within the article and its Supplementary Information files or are available from the authors upon request.

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

## Acknowledgements

We thank Alistair Siebert and Dan Clare, and Rishi Matadeen for valuable assistance in cryo-EM data collection, respectively, at eBIC Diamond (Harwell—UK) and at NeCEN (Leiden—Netherlands). We are grateful to Sari Korhonen and Helin Veskiväli (Bio-complex unit, University of Helsinki) for skillful support in virus production and Iraide Escobes (Proteomics Platform—CIC bioGUNE) for technical help in mass-spec sample preparation. Rafael Larrosa and Darío Guerrero are greatly thanked for the support provided with the high-performance computing Picasso node at the Bio-Innovation Building of the University of Malaga, member of the Spanish Supercomputing Network. Donatello Castellana (CIC bioGUNE) is acknowledged for thoughtful discussion. This study was supported by Academy Professor (Academy of Finland) grants 255342, 256518, and 283072 (to D.H.B.), the Spanish *Ministerio de Economía y Competitividad* (MINECO/FEDER BFU2015-64541-R to N.G.A.A.), the *Red Española de Super-computacón* (BCV-2015-2-0017, BCV-2015-3-0003 to N.G.A.A.) and by the Basque *Departamento de Educación, Política Lingüística y Cultura* (Ref: PRE_2016_2_0151 to I. S.-P. and N.G.A.A.). The authors acknowledge the use of Instruct-FI and Instruct-HiLIFE Biocomplex unit (EU ESFRI Instruct Centre for Virus Production, ICVIR 2009–2017) funded by University of Helsinki and the Academy of Finland (grant 1306833). We thank MINECO for the Severo Ochoa Excellence Accreditation to the CIC bioGUNE (SEV-2016-0644). This work has been fostered by iNEXT, project number 653706, funded by the Horizon 2020 program of the European Union. We also acknowledge Diamond Light Source for access and support of the Cryo-EM facilities at the UK national electron bio-imaging centre (eBIC) (proposal EM17171), funded by the Wellcome Trust, MRC, and BBSRC.

## Author contributions

H.M.O., D.H.B., and N.G.A.A. conceived the study. I.S.-P., D.C., and H.M.O. produced the virus. D.G.-C. and N.G.A.A. collected EM data. I.S.-P., D.C., and N.G.A.A. performed EM processing. I.S-P., D.C., and N.G.A.A built and refined the atomic models. M.A. and F.E. carried out mass-spec experiments. All the authors critically contributed to the analysis and interpretation of the data. N.G.A.A. wrote the paper with contributions from the rest of the authors.

## Additional information

**Competing interests:** The authors declare no competing interests.

