## [Peer Review File · Nature Communications]

Reviewers' Comments:

Reviewer #1:

Remarks to the Author:

This manuscript presents cryo-EM studies of two archaeal halophilic viruses, HCIV-1 and HHIV-2. The low resolution cryo-EM reconstruction of HHIV-2 was reported by Abrescia and colleagues in 2015. Taking advantage of direct electron detectors in the cryo-EM field, now Abrescia and colleagues are able to reach near-atomic resolution for the capsid region of HHIV2 and a similar virus HCIV-1, and then build atomic models for the MCPs and pentons. In addition to that, they observed clear densities for proteins (GPS-II and GPS-III) beneath the morphologically distinct 2- and 3-tower capsomers, which allowed them to trace the Ca of those two proteins. This certainly provides new insights for the community interested in virus assembly mechanisms. The structural studies appear to be quite sound. Other than (6) below, I only had minor points that need to be addressed.

(1) About resolution:

A) In supplementary figure 1, what is the green line? In panels a and b, it is inconceivable that the red line is the half-maps FSC with masked capsid. One would never observe such a steep curve.

B) It is known that the half-maps FSC can be a poor estimator of cryo-EM resolution. For all the atomic models built, the authors need to provide a model:map FSC and d99 (Afonine et al 2018, Acta Crystallographica Section D).

(2) In Fig. 3, panels c and d are mislabeled.

(3) In supp fig5, some of the models from the I-TASSER server are apparently not reliable. As mentioned by the authors, the confidence scores of predicted models are from -5 to 2, the higher the better. Yet the models generated by authors are scored -4.12, -4.92 and -1.15. Apparently the first two protein models are predicted with very low confidence, and those structures should be removed, otherwise it may mislead the readers. For the VP2 with score -1.15, what is the template structure and how similar is it compared with vp2 at the sequence level?

(4) The authors talked about the vertex complexes of the two viruses. Can you actually see those complexes in the raw cryo-EM micrographs? It will be helpful to include a raw micrograph figure to show those complexes by EM.

(5) Unlike crystallography, the map threshold in cryo-EM is an arbitrary value. Unless you are doing comparison with different regions like the MCP region, mentioning this in the paper is not helpful.

(6) The GPS-II and GPS-III protein structures are some of the highlights of this paper to help understand virus assembly. However, the authors only showed the main chain trace and secondary structure of those two proteins, so it is unclear to me what they really are. Were the authors able to identify their sequences among the viron proteins? Did they do MS/MS analysis and find candidates whose predicted secondary structures match the observed secondary structures of GPS-II and GPS-III? I believe this should be included in the paper.

Reviewer #2:

Remarks to the Author:

Santos-Pérez et al. present the cryo-EM structures of the major capsids of the Haloarcula californiae virus 1 (HCIV-1) and the Haloarcula hispanica virus 2 (HHIV-2) at about 4 angstrom resolution. Two major capsid proteins (MCPs) of the two viruses share high sequence identities (70-80%). It is

reasonable that their MCP structures are almost identical. It is interesting that the two viruses have distinctly different fivefold vertex complexes. The authors claim that they built de novo the atomic models of the two MCPs VP4 and VP7 based on the cryo-EM structures. The MCP heterodimer VP4-VP7 structurally resembles the MCP of Pseudoalteromonas phage PM2 belonging to the PRD1-adenovirus structural lineage. These MCPs assemble into two types of hetero-hexamers, which have twofold and threefold symmetries respectively. The different proteins (so called GPS-II and GPS-III) underneath the twofold and threefold hexamers are not so well resolved that only backbones are traced. The authors suggest that these GPS proteins direct as well as staple the MCPs into the hetero-hexamers and to interact with transmembrane proteins on the viral internal membrane that encapsulates the dsDNA genome. The assembly of the capsid, which is assisted by the transmembrane proteins on the internal membrane, starts at the fivefold vertex (VP9), followed by the attachment of the hetero-hexamers. The manuscript will be of specific interest to readers who study viruses in PRD1-adenovirus structural lineage as well as other structural virologists. However, there are several major issues in the manuscript that should be addressed before the paper can be accepted.

Major points:

P3, L53-56: In a cryo-EM density map at ~4 angstrom resolution, some main chains might be broken and many side chains might not be resolved. How did the authors guarantee the accurate amino acid registration of their models based on the cryo-EM density maps at ~4.2 angstrom resolutions? The authors could try to average the maps of 12 copies of VP4 and the 15 copies of VP7 within the asymmetric unit respectively to improve their resolutions.

P4, L78-79: Do all "GPS-III" proteins in the 3 three-tower capsomers in the asymmetric unit have the identical conformation?

P4, L88-89: Could the twofold symmetry be caused by the random binding of the GPS-II protein to the two-tower capsomer in two orientations? The two-tower capsomer is twofold symmetrical.

P5, L94: How did the authors identify the penton protein as VP9?

P5, L103: What does "its" stand for here?

P6, L134: How did the authors identify the penton protein as VP13?

What do the green lines represent in Supplementary Fig. 1?

Minor points:

P2, L42: The abbreviation "Nos" should be defined in the main text rather than in the figure legend.

P5, 107-109: Except for the vertex complexes, are there other differences between the HCIV-1 and HHIV-2?

Reviewer #3:

Remarks to the Author:

The viruses in PRD1-adenovirus viral lineage use trimers of major capsid protein with double jelly roll fold to form hexamer and use five minor capsid proteins with single jellyroll structure to form pentamer. The icosahedral capsid of these viruses is formed by 12 pentamers and hundreds or thousands hexamers which depending on the T numbers. This double jelly-roll major capsid protein was widely

found in plant RNA virus, such as cowpea mosaic virus and PBCV-1, Bacterial phage, such as PRD1 and PM2, archaeal virus STIV, vaccinia virus and even "virophage" Sputnik. Here, Santos-Perez et al., solved two membrane containing archaeal viruses at resolutions of 4.2 and 4.4 Å, respectively. The major capsid proteins of these viruses are single jelly-roll structures. The results showed that the vp4 and vp7 interacted with each other to form the building block. The so called "GPS" protein III helps to form "three-tower" hexamer with three building blocks. "GPS" protein II connects with two building blocks and two individual VP7 to form "two-tower" hexamer. The pentamer is formed by five vp9 proteins with single jelly-roll structure. 12 pentamers and 270 hexamers together form this T = 28 icosahedral virus. The pentamer connects with membrane protein to interact with the inner viral membrane. The two viruses have distinct vertex complexes connecting the pentamer. This structure shows an interesting story about how single jelly roll protein can be organized as major capsid protein to form an icosahedral virus capsid. The evidence of "GPS" proteins is convincing. However, I do have some minor points,

Minor points

1. The 4.0 and 4.2 Å maps were used to build the atomic model for major capsid proteins and minor capsid proteins. At such a resolution, it is easy to trace the main chain of the protein. However, the registration of amino acids is very hard. Therefore, it is better to show the density map together with atomic model when is describing amino acid-amino acid interactions (such as Page 3 line 55).
2. The micrographs were collected using Falcon camera with 58,000X or 60,000X magnifications. In such large magnifications, there are normally very small anisotropic magnification problems. However, since the authors are working on large viruses with diameter larger than 70 nm (I guess), it is better to pay attention to this problem. Sometimes, it may be the fact that limit the resolution of a reconstruction of a large virus.
3. As mentioned in the method section, ~ 3000 micrographs has been collected for each virus and ~ 15, 000 thousand particles were extracted for HHV2. However, the final resolution is only about 4.4 Å. It indicates that the virus is flexible or you have some other problems in the reconstruction. It is better to try some other softwares, such as the recent published "block-based reconstruction method". I think it may help to come over the concerns in this or above minor points and improve the resolution.
4. Empty viruses have been found with 15% membrane's volume shrinkage compared with that of full virus. It has been suggested by authors that the empty virus are procapsid. It indicates that this virus has a special vertex to package viral genome. Do you have any hint in the reconstruction? Or do you have any idea about which viral genes may associated with this special vertex that helps the genome packaging?
5. These viruses used two GPS proteins to help to form two distinct hexamers. However, it is unusual that both viruses do not have any minor protein that connects the neighboring hexamers. As far as I know, such minor proteins are existed almost in all other double jelly-roll viruses. It may be interesting to compare the interactions between neighboring hexamers in these two viruses and other double jelly-roll viruses to answer the question that why double jelly-roll virus need this kind of minor capsid proteins.

January 7th, 2019

POINT-BY-POINT RESPONSE LETTER TO REVIEWERS

Blue denotes responses to reviewer comments or questions.

Red denotes changes that have been made to the text or figures.

Reviewer #1 (Remarks to the Author):

This manuscript presents cryo-EM studies of two archaeal halophilic viruses, HCIV-1 and HHIV-2. The low resolution cryo-EM reconstruction of HHIV-2 was reported by Abrescia and colleagues in 2015. Taking advantage of direct electron detectors in the cryo-EM field, now Abrescia and colleagues are able to reach near-atomic resolution for the capsid region of HHIV2 and a similar virus HCIV-1, and then build atomic models for the MCPs and pentons. In addition to that, they observed clear densities for proteins (GPS-II and GPS-III) beneath the morphologically distinct 2- and 3-tower capsomers, which allowed them to trace the C α of those two proteins. This certainly provides new insights for the community interested in virus assembly mechanisms. The structural studies appear to be quite sound. Other than (6) below, I only had minor points that need to be addressed.

(1) About resolution:

A) In supplementary figure 1, what is the green line? In panels a and b, it is inconceivable that the red line is the half-maps FSC with masked capsid. One would never observe such a steep curve.

We are sorry for the confusion generated and for not including the corresponding legend for the green line. Now the figure legend has been corrected and each coloured FSC curve explained; the green line is the resulting FSC from unmasked half maps.

Please see revised Supplementary Figure 1, page 3 in SI.

B) It is known that the half-maps FSC can be a poor estimator of cryo-EM resolution. For all the atomic models built, the authors need to provide a model:map FSC and d₉₉ (Afonine et al 2018, Acta Crystallographica Section D).

We thank the Reviewer for pointing this out. We have now included a new supplementary figure (revised Supplementary Figure 2) in which we show the model:map FSC for each of the Icosahedral Asymmetric Unit (IAU) corresponding to the HCIV-1 and HHIV-2 viruses. We have also derived the resolution estimation with the d₉₉ metric for the EM density

corresponding to the IAU, and reported both in the Supplementary Tables 1-2. Previous and current estimators are all in line with the visual inspection of the submitted EM maps.

Please see revised Supplementary Figure 2 and Supplementary Tables 1-2, respectively at pages 4 and 15-16 in SI.

(2) *In Fig. 3, panels c and d are mislabeled.*

The labelling in the corresponding figure legend has been corrected.

(3) *In supp fig5, some of the models from the I-TASSER server are apparently not reliable. As mentioned by the authors, the confidence scores of predicted models are from -5 to 2, the higher the better. Yet the models generated by authors are scored -4.12, -4.92 and -1.15. Apparently the first two protein models are predicted with very low confidence, and those structures should be removed, otherwise it may mislead the readers. For the VP2 with score -1.15, what is the template structure and how similar is it compared with vp2 at the sequence level?*

The two homology models corresponding to VP6 and VP3 have been removed accordingly. We have added the sequence similarity of the plausible I-TASSER hit for VP2 and the corresponding modelled structure.

Please see revised Supplementary Figure 7 and corresponding legend (pages 9-10 in SI).

(4) *The authors talked about the vertex complexes of the two viruses. Can you actually see those complexes in the raw cryo-EM micrographs? It will be helpful to include a raw micrograph figure to show those complexes by EM.*

We can indeed see the vertex complexes in the raw micrographs.

We have now added a further Supplementary Figure numbered as 6 in the revised version (page 8 in SI), in which we show a raw image of each virus with insets highlighting the spike complexes.

(5) *Unlike crystallography, the map threshold in cryo-EM is an arbitrary value. Unless you are doing comparison with different regions like the MCP region, mentioning this in the paper is not helpful.*

We have removed the threshold value from the figure legends.

(6) *The GPS-II and GPS-III protein structures are some of the highlights of this paper to help understand virus assembly. However, the authors only showed the main chain trace and secondary structure of those two proteins, so it is unclear to me what they really are. Were the authors able to identify their sequences among the viron proteins? Did they do MS/MS analysis and find candidates whose predicted secondary structures match the observed secondary structures of GPS-II and GPS-III? I believe this should be included in the paper.*

The Reviewer has a point here. We were indeed able to model the main chain trace and secondary structure elements at best in both GPS proteins (the resolution decreases below the capsid) but we failed to unequivocally identify their sequence among the virion proteins. However we had in mind possible candidates – VP10 and VP12 - based on the following marginal evidence:

- 1) Previous genetic and biochemical studies showing that the two proteins are the major membrane associated proteins in both HCIV-1 and HHIV-2 and that especially VP12 proteins share high sequence similarity (HCIV-2 VP10 and HHIV-2 VP10 share ~57% similarity; HCIV-2 VP12 and HHIV-2 VP12 share 93% similarity) - see Jaakkola et al., 2012 J. Virol; Demina et al., 2016 mBio);
- 2) nano-LC-MS/MS analyses of protein complexes resolved in native gel performed in this study;
- 3) and secondary structure predictions (see additional information only for Reviewers at the end of this document).

Although we deliberately did not emphasize this possibility in the original manuscript in the light of the Reviewer's suggestion we are now confident to extend this discussion in the revised version.

The sentence has been now re-phrased in the manuscript at pages 6-7 lines 198-223:

'Nano-LC-MS/MS detected VP10 and VP12 (Fig. 3c) and previous genetic and biochemical studies^{16,17} showed that the two proteins are the major membrane associated proteins in both HCIV-1 and HHIV-2. VP10 and VP12 proteins share high sequence similarity across the two viruses (HCIV-2 VP10 and HHIV-2 VP10: ~57% similarity; HCIV-2 VP12 and HHIV-2 VP12: ~93% similarity)¹⁴ and secondary structure (PSSpred in I-TASSER¹⁸) and transmembrane (TMHMM v. 1.0, <http://www.cbs.dtu.dk/services/TMHMM/>) predictions confidently indicate VP10 with high α -helical content and possessing a likely C-terminal

transmembrane helix whereas VP12 being composed of two transmembrane helices (aa 13-35 and 50-68). These observations support VP10 as candidate for the GPS-II proteins (Fig. 3c).'

Reviewer #2 (Remarks to the Author):

Santos-Pérez et al. present the cryo-EM structures of the major capsids of the Haloarcula californiae virus 1 (HCIV-1) and the Haloarcula hispanica virus 2 (HHIV-2) at about 4 angstrom resolution. Two major capsid proteins (MCPs) of the two viruses share high sequence identities (70-80%). It is reasonable that their MCP structures are almost identical. It is interesting that the two viruses have distinctly different fivefold vertex complexes. The authors claim that they built de novo the atomic models of the two MCPs VP4 and VP7 based on the cryo-EM structures. The MCP heterodimer VP4-VP7 structurally resembles the MCP of Pseudoalteromonas phage PM2 belonging to the PRD1-adenovirus structural lineage. These MCPs assemble into two types of hetero-hexamers, which have twofold and threefold symmetries respectively. The different proteins (so called GPS-II and GPS-III) underneath the twofold and threefold hexamers are not so well resolved that only backbones are traced. The authors suggest that these GPS proteins direct as well as staple the MCPs into the hetero-hexamers and to interact with transmembrane proteins on the viral internal membrane that encapsulates the dsDNA genome. The assembly of the capsid, which is assisted by the transmembrane proteins on the internal membrane, starts at the fivefold vertex (VP9), followed by the attachment of the hetero-hexamers. The manuscript will be of specific interest to readers who study viruses in PRD1-adenovirus structural lineage as well as other structural virologists. However, there are several major issues in the manuscript that should be addressed before the paper can be accepted.

Major points:

P3, L53-56: In a cryo-EM density map at ~4 angstrom resolution, some main chains might be broken and many side chains might not be resolved. How did the authors guarantee the accurate amino acid registration of their models based on the cryo-EM density maps at ~4.2 angstrom resolutions? The authors could try to average the maps of 12 copies of VP4 and the 15 copies of VP7 within the asymmetric unit respectively to improve their resolutions.

We delivered two maps: one at 4.2 Å (HCIV-1) and the other at 4.4 Å (HHIV-2). In the region of 4 Å resolution, map interpretation can be challenging, but in our case the quality of the two maps with the several large side chains enormously helped the interpretation and the sequence registering. We started the modelling in the 4.2 Å which appeared of higher quality and the C α backbones for VP4 and VP7 were built aided by cross-comparing the average maps of VP4 and VP7 (across the corresponding subunits of the IAU) and the individual densities of VP4 and VP7 proximal to the five-fold.

We have now included this information in the Methods section; please see revised text at page 13, lines 399-401

However, in light of comments of Reviewer-3 we have now improved the resolution of HCIV-1 to 3.7 Å and HHIV-2 to 3.8 Å, respectively; some of the side-chains are now better defined (please see as example the included Figure only for Reviewers at the end of this document). This has also led to a slight improvement of the quality of the corresponding models apart from further confirming the correctness of the sequence registering.

Please see revised Supplementary Tables 1-2 at pages 15-16 in SI. All stereoview insets showing the density have now been updated using the improved maps.

We would also be happy to share the final maps and models with the Reviewer before publication, if necessary.

P4, L78-79: Do all "GPS-III" proteins in the 3 three-tower capsomers in the asymmetric unit have the identical conformation?

We are sorry not to have been clear on this point. In HCIV-1 the density corresponding to GPS-III protein under the capsomer 3 is the strongest, and it has been used to trace the GPS-III backbone while the density corresponding to GPS-III proteins under capsomers 2 and 1 are weaker and we cannot state unequivocally that they possess identical conformations. However, as we have determined the structure of the related HHIV-2 virus - the GPS-III under capsomer 3 and 2 display the same conformation and are identically oriented relative to the above capsomers. Therefore we infer that GPS-III proteins within the IAU adopt a similar fold in this type of viruses.

We have now clarified this issue within the revised text at page 4, lines 103-110:

'The quality of the fitting of the HCIV-1 GPS-III model into the corresponding HHIV-2 density implies that the proteins are the same. Moreover, in HHIV-2, the density corresponding to GPS-III protein under capsomer 2 is interpretable with an equivalent model (Supplementary Fig. 5). This observation supports that the GPS-III proteins adopt the same fold although the higher flexibility and/or weaker linkage to the membrane as the membrane curvature increases towards the five-fold poles might weaken their density closer to the apices.'

P4, L88-89: Could the twofold symmetry be caused by the random binding of the GPS-II protein to the two-tower capsomer in two orientations? The two-tower capsomer is twofold

symmetrical.

The Reviewer raises the question of whether the twofold symmetry detected in the GPS-II protein is a consequence of the two-tower capsomer possessing twofold symmetry. This point has some weight - we are sorry to have been unclear on this concept.

The formation of the two-tower capsomers is, in our view, subordinate to the pre-existence of (pseudo)-dimeric GPS-II proteins on the membrane vesicle - supporting this view is the fact that one of the GPS-II sits exactly on the icosahedral twofold symmetry axis and that both GPS-II are close to the membrane vesicle and possibly anchored to it. However, we do not know whether the GPS-II is one single (or multiple) protein chain(s) displaying a pseudo-two fold symmetry or simply a homodimer.

The scenario of pre-formed/existing two-tower capsomer imposing twofold symmetry onto the GPS-II protein(s) would require a mechanism for the 'non-assisted' formation of the two-tower capsomers for which there is no structural or biochemical evidence. Indeed biochemical dissociation of the HHV-2 capsid does not lead to the release of homogeneous two- and/or three-tower capsomers in solution – see Gil-Carton et al. Structure 2015 (we have similar results for HCIV-1, *unpublished data*) – as opposed to what has been seen in vertical double beta barrel viruses such as, for example, PRD1 and/or PM2 (Benson et al. Cell, 1999; Abrescia et al. Acta Crystallogr F61, 2005).

P5, L94: How did the authors identify the penton protein as VP9?

The EM density corresponding to the penton in HCIV-1 showed clearly large side-chain residues along the traced main-chain. As in Demina et al. Virus 2017, VP9 was suggested to be a minor protein composing the capsid (please see corresponding Figure 3C of the Demina's article), we confronted the VP9 primary sequence with the location of large side chain within the density. The quality of the map allowed us to unequivocally register the VP9 sequence into the map as shown in the revised Fig. 4a inset.

We have now clarified the sentence at page 5, lines 132-134 (revised manuscript):

'As previous biochemical data^{4,9} proposed VP9 to be a minor protein composing the HCIV-1 capsid, we confronted its primary sequence with the location of the large sidechains displayed by the penton density. We structurally identified VP9...'

P5, L103: What does "its" stand for here?

This has now been clarified.

Please see revised page 5, lines 147-149:

'In HHIV-2 the penton protein was identified by gene locus (gene 22) and the model built accordingly (see Methods); the density below the penton is weaker, indicating a flexibility of the VP9's first forty residues higher than its counterpart in HCIV-1.'

P6, L134: How did the authors identify the penton protein as VP13?

The original main-text at page 5, lines 99 – 102 stated:

'This density is compatible with five spanning transmembrane helices plausibly belonging to VP13 based on nano-LC-MS/MS analysis of soluble subassemblies Fig. 3b-c); a structure also used by Sulfolobus virus STIV with a double β -barrel MCP¹.

Because of the Reviewer's comment further down in the main-text referring to the identification of VP13, we have changed the above sentence to clarify the text.

The revised sentence, at page 5 lines 140–146 now states:

'This density is compatible with five spanning transmembrane helices – a structure also used by Sulfolobus virus STIV with a double β -barrel MCP¹ - plausibly belonging to five copies of protein VP13 (MW = 8.8 kDa). VP13 was detected by nano-LC-MS/MS analysis of soluble subassemblies clustering with proteins composing the vertex complex (Fig. 4b-c); its secondary structure prediction (see Methods) suggested VP13 possessing a single transmembrane helix and compatible with the density beneath the penton.'

What do the green lines represent in Supplementary Fig. 1?

We apologize for the confusion generated with the lack of explanation of the 'green line' in the FSC graphs and the misleading curve colouring.

Now this has been fully clarified in the revised Supplementary Figure 1 and corresponding legend.

Minor points:

P2, L42: The abbreviation "Nos" should be defined in the main text rather than in the figure legend.

This has now been defined in the main-text.

See revised text at page 2, line 42.

P5, 107-109: Except for the vertex complexes, are there other differences between the HCIV-1 and HHIV-2?

From the structural point of view the vertex complexes – plausibly containing the receptor binding protein – display the most remarkable structural differences.

As mentioned within the main-text (revised) page 4, lines 102-105, we noticed differences in signal strength for the different GPS-II and GPS-III proteins across the two viruses and we see a small polypeptide chain (about 18 residues) beneath the peripentonal VP4 subunit in HCIV-1 that we do not clearly see in HHIV-2 – but apart from the above no other striking differences are detectable in the icosahedrally averaged maps.

Reviewer #3 (Remarks to the Author):

The viruses in PRD1-adenovirus lineage use trimers of major capsid protein with double jelly roll fold to form hexamer and use five minor capsid proteins with single jellyroll structure to form pentamer. The icosahedral capsid of these viruses is formed by 12 pentamers and hundreds or thousands hexamers which depending on the T numbers. This double jelly-roll major capsid protein was widely found in plant RNA virus, such as cowpea mosaic virus and PBCV-1, Bacterial phage, such as PRD1 and PM2, archaeal virus STIV, vaccinia virus and even “virophage” Sputnik. Here, Santos-Perez et al., solved two membrane containing archaeal viruses at resolutions of 4.2 and 4.4 Å, respectively. The major capsid proteins of these viruses are single jelly-roll structures. The results showed that the vp4 and vp7 interacted with each other to form the building block. The so called “GPS” protein III helps to form “three-tower” hexamer with three building blocks. “GPS” protein II connects with two building blocks and two individual VP7 to form “two-tower” hexamer. The pentamer is formed by five vp9 proteins with single jelly-roll structure. 12 pentamers and 270 hexamers together form this T = 28 icosahedral virus. The pentamer connects with membrane protein to interact with the inner viral membrane. The two viruses have distinct vertex complexes connecting the pentamer. This structure shows an interesting story about how single jelly roll protein can be organized as major capsid protein to form an icosahedral virus capsid. The evidence of “GPS” proteins is convincing. However, I do have some minor points,

Minor points

1. The 4.0 and 4.2 Å maps were used to build the atomic model for major capsid proteins and minor capsid proteins. At such a resolution, it is easy to trace the main chain of the protein. However, the registration of amino acids is very hard. Therefore, it is better to show the density map together with atomic model when is describing amino acid-amino acid interactions (such as Page 3 line 55).

We agree. The panel c in the original Fig. 1 has now become Fig. 2 in the revised version.

In this new revised version of Fig. 2, more visual information has been provided in terms of atomic interactions and charge distribution on VP4 and VP7 MCPs. In particular, insets have been added showing the described interactions with corresponding density and models (new map and re-refined model have been used).

2. The micrographs were collected using Falcon camera with 58,000X or 60,000X

magnifications. In such large magnifications, there are normally very small anisotropic magnification problems. However, since the authors are working on large viruses with diameter larger than 70 nm (I guess), it is better to pay attention to this problem. Sometimes, it may be the fact that limit the resolution of a reconstruction of a large virus.

We thank the Reviewer for pointing this out. We have read with attention the two papers that deal with anisotropic magnification correction Grant et al. 2015 and Yu et al. JSB 2016.

Implementation of Grant et al., 2015 is best performed on collected super-resolution data which is not our case, whereas implementation of Yu et al., 2016 methodology requires re-processing of the whole dataset with a software other than the originally used RELION (which doesn't correct for anisotropic magnification yet). Although this is possible, it would imply a re-start of the whole study which with the now improved corresponding EM maps for HCIV-1 and HHVI-2 to 3.74 Å and 3.78 Å resolution, respectively (thanks also to the suggestion made by this Reviewer on correcting for the Ewald sphere effect, see below) we feel that – as at this stage the improvement of this correction on our data is not assured - it might jeopardize the timeline of this work.

We will assess a possible influence of the anisotropic magnification on our data in a follow-up more methodological contribution.

3. As mentioned in the method section, ~ 3000 micrographs has been collected for each virus and ~ 15, 000 thousand particles were extracted for HHIV2. However, the final resolution is only about 4.4 Å. It indicates that the virus is flexible or you have some other problems in the reconstruction. It is better to try some other softwares, such as the recent published “block-based reconstruction method” . I think it may help to come over the concerns in this or above minor points and improve the resolution.

As mentioned in the previous point we have taken on board the Reviewer's suggestion. The 'block-based reconstruction method' which recapitulates the correction of the Ewald sphere effect on large particles was published in April 2018 (Zhu et al., Nature Communications, 2018); this type of correction was also implemented in the new released version 3 of RELION in September 2018 (Zivanov et al. Elife, 2018).

In light of the Reviewer's comment and of this new implementation in RELION v3 (we have been using RELION v1.4 and v2.0 for the processing) we have revisited our processing not only for the HHIV-2 virus at 4.4 Å resolution but also for the HCIV-1 virus at 4.2 Å.

In the case of HHIV-2, applying the Ewald sphere correction and threshold masking in the post-processing routine has led to an improvement from 4.4 to 3.78 Å resolution whereas for

HCIV-1 the same workflow map improves the final map from 4.2 to 3.74 Å resolution (see new FSC in revised Supplementary Fig. 1). We have also included a figure only for Reviewers in which we compare the previous and current maps (for HHIV-2) showing the clear improvement of the density at 3.78 Å for certain side chains in the capsid region (this figure is attached at the end of this document).

This information has now been added in the Methods section at page 10, lines 305-307 and at page 11, lines 330-332, respectively. Consequently, the previous atomic models have been re-refined against the corresponding improved EM maps and show an improved geometry (see revised Supplementary Tables 1-2) - corresponding stereoview insets showing details of the density have now been updated.

Finally, because of the relevance of the above articles, they have both been included in the reference section (new references 24 and 25).

4. Empty viruses have been found with 15% membrane 's volume shrinkage compared with that of full virus. It has been suggested by authors that the empty virus are procapsid. It indicates that this virus has a special vertex to package viral genome. Do you have any hint in the reconstruction? Or do you have any idea about which viral genes may associated with this special vertex that helps the genome packaging?

HCIV-1 and HHIV-2 both have genes encoding putative packaging ATPases (HCIV-1 ORF13 product; HHIV-2 ORF7). In addition, the type of their genomes (linear molecules) supports the idea that there could be a procapsid stage which is used for packaging through a single vertex. These genes are conserved among HHIV-2, HCIV-1, SH1, and PH1 (~95% amino acid similarity). SNJ1 with a circular genome has a more diverse putative packaging ATPase. Amino acid similarity between the HCIV-1 and SNJ1 ATPases is ~37%.

We expect that the packaging ATPase would be located at the special vertex based on the knowledge coming from PRD1 (Hong et al., *PLoS Biology*, 2014) and we are convinced that there must be a membrane-anchor for the portal made of some small transmembrane proteins. Structurally, during localised reconstruction of the vertex complex of HHIV-2 we have detected pentons lacking the pentameric spikes protrusions (see main-text page 11, line 366) that we named 'empty pentons', about 36% of the 137,352 spikes extracted. The unique vertex might be 'hidden' within this subset and classification of these spikeless vertices masking different regions along the five-fold axis has been embraced. Currently we are working on it using also other strategies; the structural search and determination of the unique vertex with the asymmetric reconstruction of this type of viruses (both HHIV-2 and HCIV-1) might constitute materials for a future publication.

We thank the Reviewer for anticipating the relevance of a such follow-up study.

5. These viruses used two GPS proteins to help to form two distinct hexamers. However, it is unusual that both viruses do not have any minor protein that connects the neighboring hexamers. As far as I know, such minor proteins are existed almost in all other double jelly-roll viruses. It may be interesting to compare the interactions between neighboring hexamers in these two viruses and other double jelly-roll viruses to answer the question that why double jelly-roll virus need this kind of minor capsid proteins.

We have re-inspected the densities for the two viruses HCIV-1 and HHIV-2 – specifically the new higher resolution maps corrected for the Ewald sphere effect – along the edges of the facet and the crevices formed by the jelly-roll towers. In neither of the two EM maps could we distinguish minor proteins gluing/cementing the neighbouring pseudo-hexameric capsomers from either the exterior or the interior of the capsid shell. Of course, this observation refers to icosahedrally ordered proteins so we cannot exclude that other proteins not obeying to icosahedral symmetry might exist.

However, in the revised manuscript we have now explicitly stated that no other minor proteins have been detected across the edges of the capsomers, page 4 lines 88-91:

‘These interactions may make up for the absence of other minor proteins cementing the pseudo-hexameric capsomers across their edges within the IAU or across the virus facets as seen in double jelly-roll viruses^{7, 12}.’

A new reference 12 has been added. Further down in the revised text we have also explained the possible reasons why these vertical single beta barrel viruses do not display cementing proteins across pseudo-hexameric capsomers, page 7 lines 230-233:

‘The absence of ordered proteins connecting neighbouring pseudo-hexamers would ease the adjustments or repositioning of the above heterodimers and monomers whose interactions would rely on interface affinities and outermost connecting loops within the MCPs (Supplementary Fig. 4).’

Yours sincerely,

INFORMATION ONLY FOR REVIEWERS

In order:

- 1) stereoview in COOT of details of previous 4.4 Å HHIV-2 cryo-EM map (red) and improved one (3.78 Å in blue) showing that the blue map displays a better definition of the side-chains (*e.g.* in VP7 MCP, chain D, residues R111 and R97).
- 2) Prediction of transmembrane helices with TMHMM of HCIV-1 proteins VP12 and VP10.
- 3) HCIV-1 VP10 secondary structure prediction using I-TASSER.

TMHMM result

HELP with output formats

```
# WEBSEQUENCE Length: 173
# WEBSEQUENCE Number of predicted TMHs: 0
# WEBSEQUENCE Exp number of AAs in TMHs: 13.41002
# WEBSEQUENCE Exp number, first 60 AAs: 0.0361
# WEBSEQUENCE Total prob of N-in: 0.21764
WEBSEQUENCE TMHMM1.0 outside 1 173
```

plot in postscript, script for making the plot in gnuplot, data for plot

VPI0

TMHMM result

HELP with output formats

```
# WEBSEQUENCE Length: 94
# WEBSEQUENCE Number of predicted TMHs: 2
# WEBSEQUENCE Exp number of AAs in TMHs: 38.26938
# WEBSEQUENCE Exp number, first 60 AAs: 30.62909
# WEBSEQUENCE Total prob of N-in: 0.86240
# WEBSEQUENCE POSSIBLE N-term signal sequence
WEBSEQUENCE TMHMM1.0 inside 1 12
WEBSEQUENCE TMHMM1.0 TMhelix 13 35
WEBSEQUENCE TMHMM1.0 outside 36 49
WEBSEQUENCE TMHMM1.0 TMhelix 50 68
WEBSEQUENCE TMHMM1.0 inside 69 94
```

plot in postscript, script for making the plot in gnuplot, data for plot

VP12

Reviewers' Comments:

Reviewer #1:

Remarks to the Author:

The authors have done a very good job addressing the concerns raised. My only remaining point is that the 3.74 Å given for the resolution should be presented as 3.7 Å, unless the authors have some argument or justification for why they have the ability to determine the resolution (poorly defined) to within a hundredth of an Ångstrom.

Reviewer #2:

Remarks to the Author:

The authors have adequately addressed my concerns.

Reviewer #3:

Remarks to the Author:

My concern has been well addressed. However, in Fig. S2, the 0.5 threshold for FSC model vs map is normally used. I have no further comment.

January 29th, 2019

RESPONSE TO THE REVIEWERS' COMMENTS:

Reviewer #1 (Remarks to the Author):

The authors have done a very good job addressing the concerns raised. My only remaining point is that the 3.74 Å given for the resolution should be presented as 3.7 Å, unless the authors have some argument or justification for why they have the ability to determine the resolution (poorly defined) to within a hundredth of an Ångstrom.

In the Abstract and Main-text we now present the resolution as 3.7 Å for the HCIV-1 and 3.8 Å for the HHIV-2.

However, as numerically RELION defines the FSC cut-off at 0.143 at 3.74 and 3.78 Å resolution for HCIV-1 and HHIV-2, respectively, we have left the two digit after the decimal point in the Methods section and in the corresponding Supplementary Figure 1.

Reviewer #2 (Remarks to the Author):

The authors have adequately addressed my concerns.

Reviewer #3 (Remarks to the Author):

My concern has been well addressed. However, in Fig. S2, the 0.5 threshold for FSC model vs map is normally used. I have no further comment.

In the revised SI version, we have also reported the FSC at 0.5 in the Supplementary Figure 2.